# Assessing the spatio-temporal risk of *Aedes*-borne arboviral diseases in non-endemic regions: The case of Northern Spain

**Bruno V. Guerrero**[1]*, **Vanessa Steindorf**[1], **Rubén Blasco-Aguado**[1], **Luís Mateus**[1], **Aitor Cevidanes**[2], **Jesús F. Barandika**[2], **Ana Ramírez de La Peciña Pérez**[3], **Joseba Bidaurrazaga Van-Dierdonck**[3], **Jesús Angel Ocio Armentia**[3], **Nico Stollenwerk**[1], **Maíra Aguiar**[1,4]*

1 BCAM - Basque Center for Applied Mathematics, Bilbao, Spain, 2 Animal Health Department, NEIKER-Basque Institute for Agricultural Research and Development, Basque Research and Technology Alliance (BRTA), Derio, Bizkaia, Spain, 3 Public Health, Basque Health Department, Bilbao, Spain, 4 Ikerbasque, Basque Foundation for Science, Bilbao, Spain

* bguerrero@bcamath.org (BVG); maguiar@bcamath.org (MA)

**Data availability statement:** Due to legal and ethical restrictions, the epidemiological dataset

## Abstract

Arboviral diseases represent a growing global health challenge. While dengue cases surge in endemic regions, non-endemic areas in southern Europe are seeing a rise in imported cases of dengue, Zika, and chikungunya, along with the first autochthonous dengue transmissions. The expanding *Aedes* mosquito populations, influenced by climate change, and increased international travel introducing viremic cases further elevate the risk of outbreaks. These trends emphasize the urgent need for effective risk assessment and timely intervention strategies. We present a data-driven methodology to assess the spatio-temporal risk of *Aedes*-borne arboviral diseases in non-endemic settings, addressing key limitations of models developed primarily for endemic regions and challenges related to limited data availability. Our approach builds on the SIRUVY human–vector compartmental model and incorporates stochastic formulations to capture variability in imported cases and mosquito density - two critical drivers of autochthonous transmission and outbreak emergence. This framework improves risk estimation and offers insights into transmission dynamics in regions where outbreaks are rare and unpredictable, shaped by sporadic case importations and a non-persistent vector presence. Using data from the Basque Country (2019–2023), including *Aedes* mosquito egg counts as a proxy for vector abundance and records of imported cases, we mapped the monthly risk of local transmission at the municipal level and conducted a scenario-based risk assessment aligned with Spain's entomological classification. Our findings indicate a growing presence of *Aedes* mosquitoes and an increasing transmission risk in urban and peri-urban areas of the Basque Country, revealing shifting hotspots of possible arboviral disease transmission. These results highlight the importance of sustained surveillance

used in this study cannot be made publicly available. The data contain potentially identifying and sensitive patient information, and access is regulated by the Department of Public Health of the Basque Country. Requests for access to the anonymized minimal dataset may be directed to: Contact: Vigilancia Epidemiológica del Departamento de Salud del Gobierno Vasco Email: p-latasa@euskadi.eus. Data on vector abundance are managed by Instituto Vasco de Investigación y Desarrollo Agrario. Requests for access the dataset may be directed to: Contact: Departamento de Sanidad Animal de NEIKER, Instituto Vasco de Investigación y Desarrollo Agrario Email: jgarrido@neiker.eus. The data are available upon reasonable request to researchers who meet the criteria for access to confidential health data and have ethics approval.

**Funding:** This research was supported by the Spanish Ministry of Science, Innovation and Universities through the BCAM Severo Ochoa accreditation (CEX2021-001142-S/MICIN/AEI/10.13039/501100011033 to MA), and by the Basque Government through the "Mathematical Modeling Applied to Health" project and the BERC 2022–2025 program (funding to MA). This work was also funded by the ARBOSKADI project for monitoring vector-borne diseases in the Basque Country, Euskadi (funding to MA). Maíra Aguiar (MA) and Aitor Cevidanes (AC) acknowledge financial support from the Spanish Ministry of Science and Innovation (MICINN) through the Ramon y Cajal grants RYC2021-031380-I (to MA) and RYC2021-033084-I (to AC). The collection of mosquito data was funded by the Department of Food, Rural Development, Agriculture and Fisheries and the Department of Health of the Basque Government, the Ministry of Health, Social Policy, and Equality of the Government of Spain, and the EU-LIFE project 18 IPC/ES/000001 (Urban Klima 2050 to AC). The funders had no role in study design, data collection and analysis, decision to publish, or preparation of the manuscript.

**Competing interests:** The authors have declared that no competing interests exist.

to identify high-risk locations and prioritize targeted public health interventions to prevent potential outbreaks.

## Author summary

Arboviral diseases such as dengue, Zika, and chikungunya are an increasing global health concern. While traditionally confined to tropical regions, these diseases are now emerging in new areas, including parts of Europe, due to rising international travel and environmental changes that support the spread and establishment of *Aedes* mosquitoes. This shift highlights the urgent need for tools to assess and manage outbreak risks in previously unaffected regions. In this study, we present a data-informed modeling framework to assess the risk of local transmission of *Aedes*-borne diseases in non-endemic areas where the mosquito vector has recently become established. We adapted a classical stochastic model of disease transmission to reflect local conditions, incorporating mosquito abundance and the arrival of viremic travelers to simulate different outbreak scenarios. By integrating real-world entomological and epidemiological data, we produced monthly risk maps for the Basque Country at the municipal level and evaluated risk under various scenarios. Our approach is flexible and can be updated with new data or adapted to different settings and diseases. It offers a valuable tool to support public health planning, improve preparedness, and guide timely interventions against emerging vector-borne threats.

## Introduction

Mosquito-borne diseases pose a significant global health threat, exacerbated by climate change and global warming. The transmission of these diseases is highly sensitive to climatic factors that affect both vector adaptability and pathogen development [1–5]. Environmental changes and socio-demographic shifts, such as deforestation and urbanization, are further contributing to the expansion of both emerging and resurgent diseases [3,6,7].

Climate change, combined with rising international human mobility and trade, not only facilitates the proliferation of disease vectors, but also drives the introduction of pathogens into naïve areas, increasing the risk of local vector-mediated transmission to susceptible populations [8–12]. As a result, neglected diseases like dengue and other mosquito-borne illnesses are spreading rapidly beyond their traditional endemic zones [13,14]. In contrast to endemic regions, where local transmission cycles are sustained, non-endemic regions primarily rely on imported viremic cases—confirmed infections acquired abroad by travelers—as the main source of infection. The establishment of invasive *Aedes* species habitats, combined with the arrival of imported cases, can lead to significant outbreaks of locally transmitted (autochthonous) infections in individuals with no recent travel history.

For example, in Europe, historically a non-endemic region for diseases like dengue, Zika, and chikungunya, local outbreaks have already occurred. These include the DENV1 outbreak in Madeira Island, Portugal, in 2012 [15], two Italian CHIKV outbreaks in 2007 and 2017 [16], and recent dengue outbreaks in France and Italy (2023 and 2024) [17]. The increasing frequency of imported and autochthonous cases underscores the growing burden of mosquito-borne diseases in European countries.

Risk assessment of mosquito-borne diseases, including prevention, management, surveillance, and communication, is a complex challenge in public health, requiring coordinated

efforts from international agencies [18]. Decision-support tools, such as Multi-Criteria Decision Analysis [19] and Risk Matrices [20–22], though used to a limited extent by public health managers and stakeholders, can help identify vulnerable areas to mosquito-borne diseases [23–26]. These tools rely on expert judgment to subjectively weigh explanatory variables and empirical factors, such as costs, perceived risks, strategic objectives, and resource allocation priorities.

Despite the lack of standardized methods and metrics for risk assessment, review articles by Sedda [18], Louis [27], Tjaden [28], and Lim [29] provide complementary summaries of commonly used approaches for mapping arbovirus transmission risk. They highlight limitations in these methods and advocate for further refinement and greater comparability. These studies generally classify models into 'correlative' species distribution models (e.g. [30]), including Bayesian hierarchical models and spatio-temporal approaches [31], and 'mechanistic' epidemiological models [32,33]. Correlative models struggle with obtaining reliable presence/absence data, translating environmental suitability into occurrence probabilities, and defining appropriate environmental predictors. Mechanistic models, on the other hand, rely on high-quality spatiotemporal data for parameter estimation and validation, often sourced from endemic regions. When applied to new areas, these models may fail to account for the adaptation of vectors to local environmental conditions, particularly at small spatial scales and short time periods.

In mathematical epidemiology, the SIRUV model framework [34,35] captures human-mosquito interactions in the transmission of vector-borne pathogens, providing insights into these dynamics, particularly for short-term predictions. While deterministic and stochastic models are not designed as direct risk estimators [36], the values of $R_0$ derived from these models can help create meaningful risk maps [37,38]. However, the magnitude of $R_0$ is not directly comparable across different models, even though all share the threshold of $R_0 = 1$. Moreover, $R_0$ does not fully account for imported cases, which play a significant role in introducing new infections to non-endemic regions.

In response to the growing concern over mosquito-borne diseases in non-endemic countries, this study presents a practical framework for spatio-temporal risk mapping, using the expected number of autochthonous cases as a risk estimator. In this study, we hypothesize that dengue outbreaks in temperate, non-endemic regions can emerge even under subcritical vector abundance conditions, primarily driven by the stochastic introduction of imported cases. Our objective is to develop a modeling framework capable of capturing this complex dynamic—one in which outbreak occurrence cannot be reliably forecast due to limited or sparse epidemiological data, but early warning signals may still indicate the likelihood of local transmission. Strengthening surveillance to detect such signals may prove critical for timely and proactive public health interventions.

Our approach is aimed to complement existing methodologies by providing a simple, interpretable, and data-driven tool for early outbreak risk assessment, particularly suited to non-endemic settings. We apply this framework to the Basque Country, Spain, where data on vector abundance and imported cases are available. The method integrates a modified SIRUV model with qualitative entomological classifications, spatially aggregated at the subnational level, and adapted to the data constraints typical of non-endemic regions. This framework allows for the identification of spatial and temporal patterns of potential local outbreaks, supporting the prioritization of higher-risk areas and facilitating ongoing surveillance. Once the relevant data are available, the framework can be readily adapted to other regions facing the emergence of *Aedes*-borne diseases.

## Materials and methods

### Geographical scope and data sources

The study focuses on the Basque Country, an autonomous community in Northern Spain. Although no autochthonous cases of tropical mosquito-borne diseases such as dengue, chikungunya, and Zika have been reported in the region, viremic imported cases are frequently observed. The invasive *Aedes* mosquito species, first detected in the Basque Country in 2014, have since become progressively established, with their populations now widespread across several municipalities [39,40].

This region spans 7,230 km$^2$ [41] and is home to 2,201,462 inhabitants [42]. Administratively, it is divided into three provinces: Gipuzkoa, Bizkaia, and Araba, which encompass 252 municipalities. Gipuzkoa borders France to the east, Bizkaia is the most densely populated and industrialized, while Araba is the largest in area and the least populated. It has Atlantic and sub-Mediterranean climates [43], with mild temperatures and significant rainfall, along with a varied topography that includes mountains, extensive forests and natural parks, its northern coastline and three major river systems.

The Basque Country, one of Spain's autonomous communities with the highest per capita income, fosters both domestic and international mobility among its residents and has also emerged as an increasingly popular destination for travelers from other parts of Spain and abroad. This dynamic movement of people, combined with the region's environmental, entomological, and geographical features, shapes its epidemiological landscape. Since 2016, the Public Health Department of the Basque Country has closely monitored the arrival of imported symptomatic viremic cases, given their potential to initiate local transmission through the established mosquito population. For this study, the epidemiological dataset comprises geo-referenced records of confirmed imported cases of dengue, Zika, and chikungunya. Each entry includes the latitude and longitude of the reporting health center, as well as the precise notification date recorded on a daily basis.

Entomological data on mosquito presence were provided by NEIKER (Basque Institute for Agricultural Research and Development). The cleaned dataset used in this study comprises biweekly records of egg counts per ovitrap stick, including geographic coordinates, collection dates, and the identified *Aedes* species. Since 2013, an annual surveillance program from June to November has deployed ovitraps in municipalities with over 10,000 inhabitants. In each sampling area, five ovitraps were installed, with oviposition sticks replaced every 15 days. Eggs were counted, and *Aedes* species were identified via laboratory analysis. Each municipality included two sampling zones, with up to ten zones in provincial capitals. During colder months (December to May), mosquito presence was not detected, as confirmed by year-round monitoring at four sentinel sites (two in Gipuzkoa and two in Bizkaia), which were established to track the seasonal activity of *Aedes* mosquitoes in previously affected regions. Further details of the entomological surveillance program can be found in [39,40,44].

The present study analyzes data from 2019, 2022, and 2023, focusing on viremic imported cases of dengue, Zika, and chikungunya (see Fig 1). Data from the COVID-19 pandemic period, covering 2020 to 2021, were excluded, as lockdown measures significantly reduced the number of imported cases. However, vector surveillance continued uninterrupted throughout this period (as shown in Fig 1). This analysis focuses on two invasive mosquito species: *Aedes albopictus* and *Aedes japonicus*, the latter recognized as a competent vector for West Nile virus and a potential vector for dengue, Zika, and chikungunya [45]. Records from other species, such as *Aedes geniculatus* and *Culex pipiens*, were excluded due to their lack of vector competence for the diseases under study.

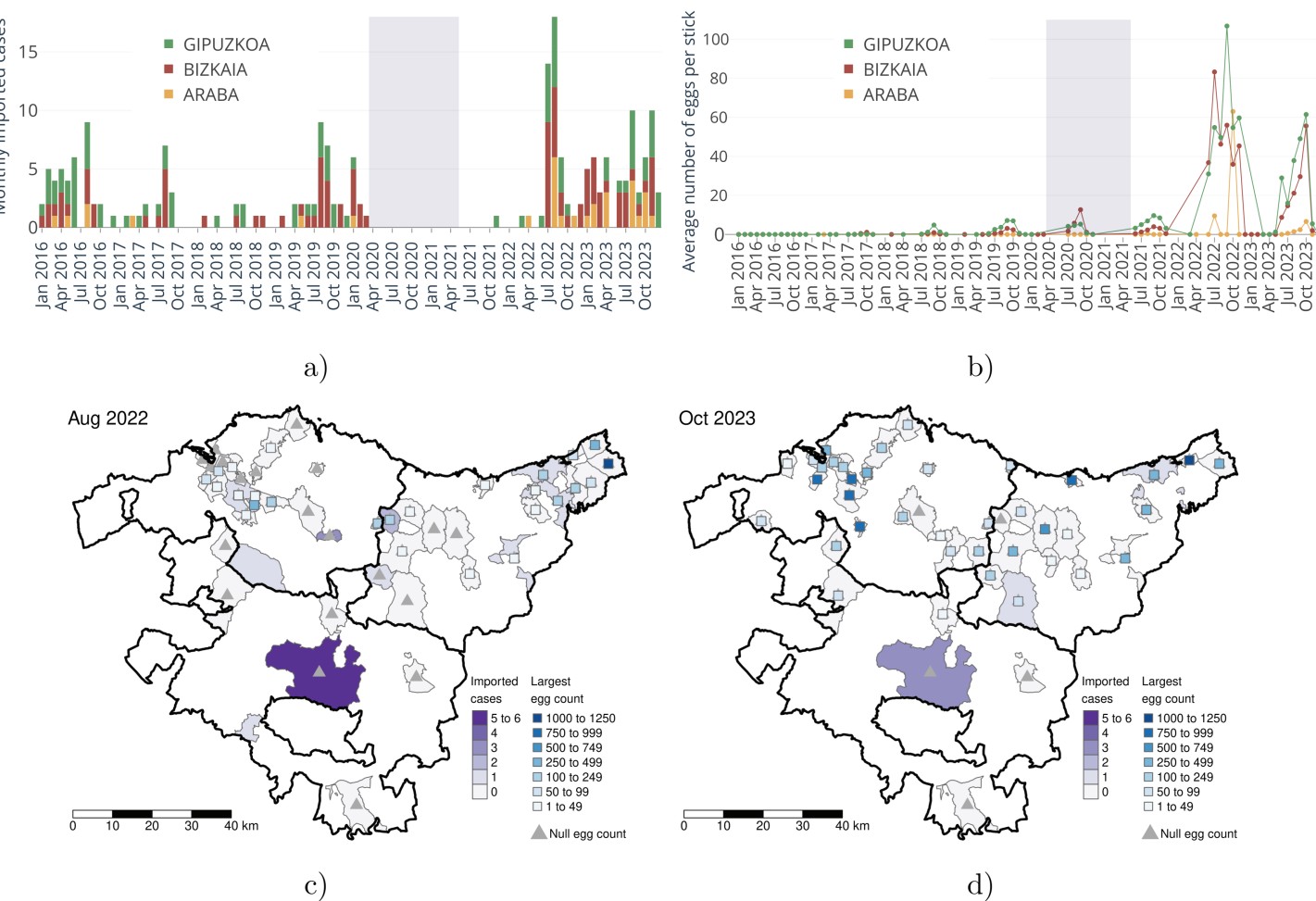

**Fig 1. Empirical data from the Basque Country used in this study.** Imported cases of mosquito-borne diseases (dengue, Zika, or chikungunya) and egg counts from mosquito ovitraps (*Aedes albopictus* and *Aedes japonicus*). In panel (a), the monthly time series of reported cases by province is shown. Panel (b) presents the average egg counts per ovitrap stick. The light purple-highlighted period corresponds to the COVID-19 lockdown, during which the number of imported cases was significantly reduced. Maps in panels (c) and (d) illustrate data aggregated by municipality for August 2022 and October 2023, the months with the highest recorded epidemiological activity. These maps focus on municipalities with reported imported cases or where vector surveillance was conducted. Ovitrap locations are marked with squares, with color indicating the highest egg count at each site. Gray areas represent municipalities with no reported cases, and gray triangles indicate ovitraps with negative results. Thicker lines represent the administrative boundaries of the Basque Country's provinces: Gipuzkoa (Northeast), Bizkaia (Northwest), and Araba (South). Monthly maps for 2019, 2022 and 2023 are available in S1 Fig. Base map layer from the Basque Government (Eusko Jaurlaritza / Gobierno Vasco) resource https://www.euskadi.eus/limites-administrativos-del-pais-vasco/web01-ejeduki/es/, under CC BY 4.0 https://creativecommons.org/licenses/by/4.0/.

## Model-based risk estimator for non-endemic regions

Built upon the SIRUV framework [34,35], which combines a Susceptible-Infected-Recovered (*SIR*) model for humans with the Uninfected-Infected Vector (*UV*) model for mosquitoes, this minimalistic model is refined to incorporate the dynamics of viremic imported cases (*Y*). This modification is essential for characterizing the population-level infectious dynamics of mosquito-borne diseases in non-endemic areas, where local transmission is expected to be primarily driven by imported cases. The SIRUVY model [46] is governed by the following

system of ordinary differential equations

$$
\begin{aligned}
\frac{d}{dt}S &= \alpha R - \frac{\beta}{mN}SV \\
\frac{d}{dt}I &= \frac{\beta}{mN}SV - \gamma I \\
\frac{d}{dt}R &= \gamma I - \alpha R \\
\frac{d}{dt}U &= \psi(k) - \frac{\vartheta}{N}U(I+Y) - \nu U \\
\frac{d}{dt}V &= \frac{\vartheta}{N}U(I+Y) - \nu V \\
\frac{d}{dt}Y &= \hat{\varrho}N - \gamma Y,
\end{aligned}
\tag{1}
$$

and the dynamics, illustrated in Fig 2, can be summarized as follows.

Susceptible humans $S$ become infected through the bites of infected mosquitoes $V$ at a rate $\beta$. Infected individuals $I$ recover from the infection at a rate $\gamma$, while recovered individuals $R$ gradually lose immunity and revert to being susceptible at a waning immunity rate $\alpha$. Note that the standard incidence [33] is used in the assumptions when constructing the model. Additionally, there is a steady external import of new infected cases $Y$, introduced at a constant arrival rate $\hat{\varrho}$.

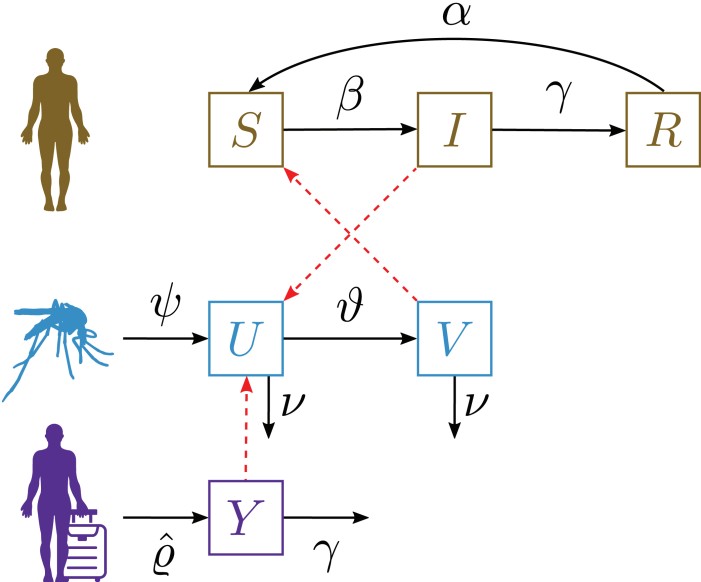

**Fig 2. Flow diagram for the SIRUVY model.** Colors represent different compartment classes and transitions: brown for local transmission in human hosts (susceptible, infected, and recovered), cyan for uninfected and infected disease vectors, and violet for the influx of confirmed imported human cases $Y$. Red dashed arrows indicate pathogen transmission pathways, while black arrows represent the flow of individuals between compartments, with the corresponding flow rates denoted by Greek letters. Icons were sourced from https://www.svgrepo.com/svg/76394/standing-human-body-silhouette, https://www.svgrepo.com/svg/95482/mosquito, https://www.svgrepo.com/svg/490438/travel-luggage, all under CC0/Public Domain license https://www.svgrepo.com/page/licensing/.

On the mosquito side, uninfected mosquitoes $U$ become infected by biting infected individuals $I$ at a rate $\vartheta$, transitioning into infected vectors $V$ that are able to transmit the disease. The dynamics of the mosquito population are driven by the rates $\psi$, which represents the influx of female *Aedes* mosquitoes, and $\nu$, which denotes their natural mortality rate.

For simplicity, the model assumes that both the human population $N$ and the mosquito population $mN$ (where $m$ is the ratio of mosquitoes to humans) remain constant, neglecting any demographic changes.

In this context, the vector supply rate $\psi$ is defined as

$$\psi(k) = k \nu\, mN\,, \tag{2}$$

where

$$k = \frac{\text{vector abundance in the non-endemic area}}{\text{vector abundance in an endemic area}} = \frac{\kappa}{\kappa_{end}}\,, \tag{3}$$

is a dimensionless parameter that quantifies mosquito abundance in the non-endemic region relative to the estimated abundance in an endemic region. An assumption in the SIRUVY model is that, when the vector is not fully established, $k$ is expected to be less than 1. To ensure comparability of $\kappa$ estimates, the same sampling protocols (e.g., ovitraps, light traps, $CO_2$ traps, gravid traps, larval sampling, or positive trap percentage [47]) should be used in both regions.

The imported viremic cases, $Y \ll N$, are modeled as independent events occurring at a constant rate $\hat{\varrho} = \varrho\gamma$ over a given time period, resulting in an average of $\langle Y \rangle = \varrho N$. Finally, the expected number of autochthonous cases can be derived from the steady-state solution of the infected compartment, where $\frac{dI}{dt} \to 0$, as follows

$$\langle I \rangle = k\beta_{\text{eff}} \frac{1}{\varepsilon(k)} \langle Y \rangle\,. \tag{4}$$

In equation (4), angular brackets denote the expected value, while

$$\beta_{\text{eff}} = \beta \frac{\vartheta}{\nu} \frac{S_0}{N}$$

represents the effective infection rate, with $S_0$ being the initial size of the susceptible population. The term

$$\varepsilon(k) = \gamma - k\beta_{\text{eff}}$$

represents the deviation from the epidemiological threshold, $k_c = \frac{\gamma}{\beta_{\text{eff}}}$, indicating the distance from the exponential growth threshold where self-sustained transmission occurs within the local population, depending on the relative mosquito abundance $k$. As $k$ approaches the critical value $k_c$, $\varepsilon(k)$ tends to zero, resulting in exponential growth in the expected number of cases.

In scenarios with low mosquito abundance ($k \to 0$), the denominator simplifies to $\varepsilon(k) \to \gamma$, thereby eliminating the nonlinear contribution of $k$. In this case, the expression for the expected number of autochthonous cases reduces to a simple linear form:

$$\langle I \rangle = \xi\, k \langle Y \rangle\,, \tag{5}$$

where the constant of proportionality $\xi := \frac{\beta_{\text{eff}}}{\varepsilon} \approx \frac{\beta}{\gamma} \frac{\vartheta}{\nu} \frac{S_0}{N} \approx 1.5$, based on the values provided in Table 1.

This expression for $\langle I \rangle$ serves as a proxy to assess the risk of autochthonous cases of arbovirus diseases in non-endemic areas and time periods, as proposed in [46]. It relies only on the available mosquito abundance (relative to endemic levels) and the average number of imported infected cases. However, while this result is derived from a deterministic model, stochasticity is essential to account for the irregular behavior and large fluctuations observed in real disease incidence data [48].

To gain deeper insights into the dynamic behaviors described by the model in non-endemic settings, Fig 3 illustrates the range of expected outcomes as the dimensionless vector abundance parameter $k$ varies from low to high values. In addition to the deterministic solution, stochastic simulations were conducted using the Gillespie algorithm [50] to explore the effects of random fluctuations on the system's behavior. The model outputs show how the system transitions from initial exponential growth at high mosquito densities to linear growth near the threshold, eventually reaching a saturated regime where $\langle I \rangle$ stabilizes.

Due to inherent stochasticity, individual realizations may deviate considerably from the deterministic solution, especially near the key epidemiological threshold $k_c$, or when the number of initial infections is low. Even when the system reaches equilibrium, fluctuations can still trigger medium to large outbreaks, sometimes after several years without reported infections. Although the average number of autochthonous cases $\langle I \rangle \ll 1$ in Fig 3(a)–3(b), a few cases forming clusters may still occur. Interestingly, even without initial infections (as shown in Fig 3(c)), stochastic simulations can still approximate deterministic outcomes, though the timing of outbreaks remains uncertain, depending on the emergence of an index case. This variability emphasizes the importance of accounting for stochastic effects in low-endemicity settings, where random fluctuations can significantly shape the trajectory of individual outbreaks and hinder accurate predictions in real-world scenarios. In non-endemic regions, where outbreaks are expected to be triggered by imported cases, $Y_0$ plays a critical role - its timing and magnitude can either accelerate or delay the onset of transmission, as illustrated in Fig 3(c).

**Table 1. Baseline parameters used to solve the SIRUVY model.**

| Rates | Description | Value ($year^{-1}$) |
|---|---|---|
| $\gamma$ | Human recovery | 52 |
| $\beta$ | Human transmission | $1.2\gamma$ |
| $\alpha$ | Human waning immunity | 1/65 |
| $\nu$ | Mosquito death | 36.5 |
| $\vartheta$ | Mosquito infection | $1.2\nu$ |
| $\psi$ | Mosquito supply* | $\nu M = \nu mN$ |
| Population sizes | Description | Value |
| $N$ | Human* | 10000 |
| $mN$ | Mosquito* | $10N$ |
| $S_0$ | Initial susceptible humans* | $N$ |
| $U_0$ | Initial uninfected mosquitoes* | $mN$ |
| $Y_0$ | Initial imported population* | 0 |
| $\varrho$ | Imported cases (population-adjusted)* | $10^{-5}$ |

Numerical values were extracted from [34,49], while those parameters marked with (*) have been adjusted to reflect values specific to the simulated non-endemic setting (see Fig 3). The value $\varrho = 10^{-5}$ results from taking the ratio of the order of magnitude of the typical monthly number of imported cases (a few units) to the total Basque population size ($10^6$).

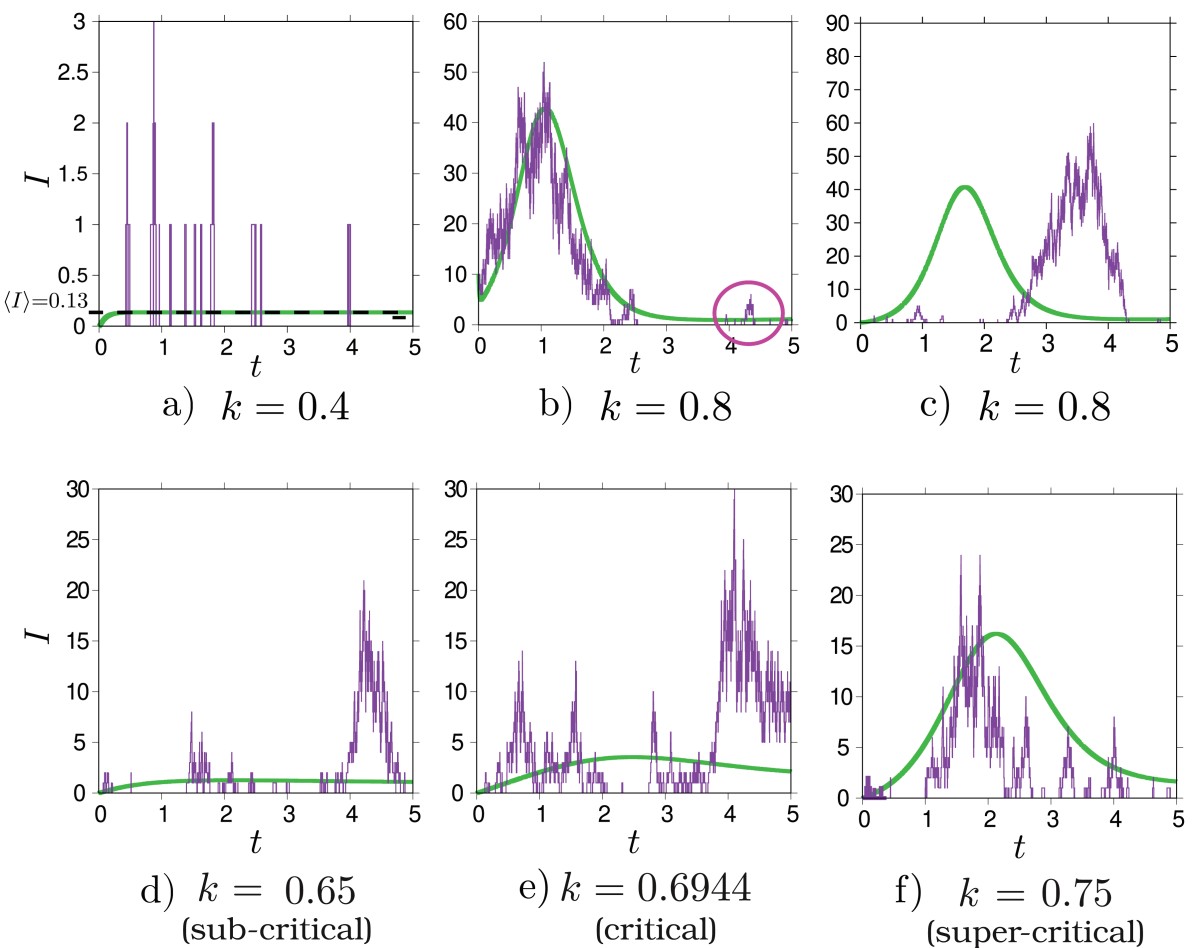

**Fig 3. Disease dynamics in non-endemic settings as a function of relative mosquito abundance *k*, with $\langle Y \rangle$ = 0.1.** Stochastic realizations (purple) are compared with the corresponding deterministic ODE solution (green). (a) At low *k*, small clusters may form despite $\langle I \rangle \ll 1$. (b) At high *k* (and $I_0$ = 10), outbreaks can re-emerge after seeming extinction (see magenta oval). (c) In absence of initial imported cases, index cases trigger medium-to-large outbreaks, occurring earlier or later than predicted by the deterministic solution. (d–f) Near the critical value ($k_c$ = 0.694), *I(t)* at the initial stage: for $k < k_c$, it quickly stabilizes at its stationary value; when $k \approx k_c$, it grows linearly with time; and for $k > k_c$, it exhibits exponential growth.

Given that any risk measure should focus on expected behaviors rather than rare occurrences, the proposed risk formula effectively addresses this need, making it a valuable tool for assessing outbreak potential. Once key parameters such as $\langle Y \rangle$ and *k* are estimated from empirical data, the stochastic framework can serve as an instrument for sensitivity analysis. This combined approach enables the evaluation of expected scenarios and how changes in model parameters resulting from hypothetical public health interventions or environmental factors can influence the risk of outbreaks, providing key insights for the design of effective surveillance, prevention, and control strategies in non-endemic regions.

## On spatio-temporal scales for risk assessment

Once the model-based risk estimator is defined, the next step is to establish the appropriate spatial and temporal scales for aggregating the gathered data. The expression in Eq (5) does

not inherently constrain a specific temporal or geographical scale; however, the way in which data is aggregated significantly impacts the measured risk.

From a pragmatic perspective, the geographical scale should be determined based on the jurisdiction of the entity responsible for implementing public health actions. For example, in Spain, if measures are enacted by public health organizations, the operational scale may be at the level of Integrated Health Systems (IHSs) or Health Zones (HSs). In contrast, if the measures are political or economic, the scale should align with provincial, county, or municipal levels, depending on the jurisdiction of the governmental entity involved.

While selecting the appropriate scale may seem straightforward, it requires careful consideration and is not arbitrary. In the Basque Country, a visual inspection of the ovitrap sampling areas (see [40]) and the georeferenced dataset of imported cases suggests that the ideal descriptive scale should not be as broad as Integrated Health Systems (IHSs) or provinces. Aggregating data over larger areas may increase the frequency of available data per location, but it could also underestimate risk in urban centers and overestimate it in less populated regions. Conversely, using very small spatial scales, such as neighborhoods, may not yield enough data. Therefore, the municipal scale is found to be optimal, with provincial-level aggregation as a fallback for municipalities with insufficient data.

Regarding the temporal scale, although epidemiological reports are provided daily, the biweekly ovitrap counts and the infrequent occurrence of imported cases (typically just a few per month) suggest that the most appropriate scale for analysis is monthly, with yearly aggregation for months where data is missing.

By using epidemiological and entomological reports, we propose a method to compute the necessary monthly (or yearly) and municipal (or provincial) data aggregations to determine $k$ and $\langle Y \rangle$.

**Determining the mosquito abundance $\kappa$.** In a non-endemic region, the abundance of mosquitoes, denoted as $\kappa$, is computed based on the ovitrap egg counts collected during the relevant time period. Instead of relying on the maximum or average egg count, we use the average of the $n$ largest egg counts per ovitrap, referred to as $\kappa_n$. This approach helps mitigate the influence of anomalous counts from individual traps, while also avoiding underestimation due to traps that consistently record low counts.

The entomological records used in this study are based on oviposition trap (ovitrap) egg counts - an established and widely used method for detecting the presence and seasonal activity of *Aedes* mosquitoes. Although ovitraps do not directly measure adult mosquito abundance [47], studies such as [51] have shown a statistically significant positive relationship between egg counts and the mean number of biting adult females. This supports their use as a practical proxy for adult female abundance, particularly when more resource-intensive adult trapping methods are not feasible. Ovitraps are especially suited to non-endemic regions with low mosquito densities, where they offer high sensitivity, minimal maintenance, and low operational costs [52], making them a valuable tool for early warning systems and targeted vector control planning.

Instead of estimating the abundance by using the highest value of egg count (which may overestimate it due to atypical high values) or by taking the average (which may underestimate it for those traps with low egg counts), we compute $\kappa_n$ using $n = 3$ for monthly aggregation at the municipal level and $n = 20$ for yearly aggregation at the provincial level.

To derive the ratio $k$, we apply Eq (3), where the calculated $\kappa_n$ is divided by the typical egg count for a trap placed in an endemic country, denoted as $\kappa_{end}$. Accurately determining $\kappa_{end}$ is challenging due to the lack of standardized egg collection protocols, which complicates direct comparisons. For simplicity, we assume $\kappa_{end} = 1000$, noting that alternative values would act

merely as scaling factors when computing $\langle I \rangle$ and do not affect relative comparisons. In this study, to preliminarily account for annual seasonality, we adjust $\kappa_{20}$ by a factor of $\frac{1}{2}$.

**Determining the expected number of imported cases $\langle Y \rangle$.** In a non-endemic setting, the random arrival of viremic imported cases can be modeled as a Poisson process, under the premise that these events occur independently at a constant effective rate $\lambda$, defined as

$$\lambda = \hat{\varrho} N = \varrho \gamma N \, . \tag{6}$$

This rate can also be directly estimated from the time series of imported case incidences using the maximum likelihood estimator

$$\hat{\lambda} = \frac{1}{\frac{1}{n} \sum_{i=1}^{n} \tau_i} \, . \tag{7}$$

Here, $\tau_i$ represents the time interval between two consecutive cases, and its confidence intervals can be estimated [46].

The expected number of imported cases for a time period $T$ is given by

$$\langle Y \rangle = \frac{C(t_{end}) - C(t_{ini})}{\gamma T} \quad , \tag{8}$$

where $C(t_{end})$ and $C(t_{ini})$ are the cumulative number of cases at the end and beginning of the time period $T$.

In terms of $\hat{\lambda}$, Eq (8) can be expressed as

$$\langle \hat{Y} \rangle = \frac{\hat{\lambda}}{\gamma} \, . \tag{9}$$

This estimation becomes more reliable as the number of events increases.

At the municipal and monthly scale, given the limited data, we apply Eq (8) to calculate $\langle Y \rangle$. Conversely, at the provincial and yearly scale, we use the statistical estimator provided by Eq (9), thanks to the better sampling available.

Limitations in case detection and reporting may lead to underestimation of the true number of infections, particularly in countries with incomplete surveillance for vector-borne diseases. Nevertheless, using statistics on the temporal distribution and waiting times of reported cases provides more robust estimates of importation than simple averages, as it captures stochastic temporal dynamics more accurately.

We model the importation process as a homogeneous Poisson process within fixed monthly intervals—rather than over the entire study duration—as a first-order approximation. This approach assumes independence between events and exponentially distributed inter-arrival times, facilitating tractable likelihood estimation under limited temporal resolution [46].

Although formal goodness-of-fit tests (e.g., Kolmogorov–Smirnov, Chi-square) could in principle assess this assumption, their power is limited by the small number of cases per window (typically ≤6; see Figs 1 and 3a). Thus, non-rejection should not be taken as confirmation, but rather as lack of evidence for a more complex alternative.

Seasonal variation in importation is expected—driven by transmission patterns in endemic areas and travel fluctuations—but the sparsity of cases makes it difficult to detect or validate such trends. More flexible models with time-varying rates would require assumptions unsupported by the available data (see Fig 3(a)).

Moreover, travelers arrive from diverse endemic regions with overlapping epidemic periods, leading to aggregated importation patterns that appear irregular and non-periodic. Even during global transmission peaks, importations into a given area may remain low or absent over short intervals.

Underreporting further complicates inference, especially in non-endemic regions with limited clinical awareness of arboviral infections. Mild or asymptomatic cases often go undetected, meaning the true number of importations likely exceeds reported figures.

Given these constraints, the constant-rate Poisson model offers a conservative yet practical approximation that balances simplicity and data-driven realism—appropriate for sparse, uncertain settings like the non-endemic regions studied here.

**Risk evaluation for the chosen spatio-temporal scales.** To ensure usability, we outline the risk expressions used in this study, tailored to the available data and the considerations discussed in this section. Table 2 summarizes the corresponding expressions for each scenario. These are applied at a municipal-monthly scale where data is accessible, or adjusted to a provincial-yearly scale when the former is not feasible.

This dual scaling at spatial (municipal/provincial) and in time (monthly/yearly) levels not only balances better resolution and practicality, but also provides a good starting point when initiating exploratory studies in the absence of prior knowledge for a new non-endemic area. This way, more effective and tailored public health interventions can be designed for each region, while ensuring broad and consistent coverage in areas with limited data.

## Results

In this section, the risk assessment for the Basque region will be quantified using the risk estimator provided in Eq (5), tailored to the spatial and temporal scales outlined in Table 2. For municipalities with sufficient data, quantities are computed at the municipal-monthly level. In cases of missing or incomplete data, provincial or yearly averages are used to supplement the estimates, providing a baseline for municipalities with limited information. This approach ensures that all regions are included in the risk assessment, even in the absence of complete data.

**Table 2. Risk formulas tailored to the available data.**

| Risk Estimator based on | Positive Egg Count | Zero/Null Egg Count |
|---|---|---|
| $\langle I \rangle = \xi k \langle Y \rangle$ | $k = \frac{\kappa_3}{\kappa_{end}}$ | $k = \frac{1}{2}\frac{\kappa_{20}}{\kappa_{end}}$ |
| $\langle Y \rangle$ when reported cases $(\frac{Y}{\gamma T})$ | $\frac{1.5}{4}\frac{\kappa_3}{\kappa_{end}} Y$ | $\frac{1.5}{2}\frac{\kappa_{20}}{\kappa_{end}}\frac{Y}{\gamma T}$ |
| $\langle Y \rangle$ when non-reported cases $(\frac{\lambda}{\gamma})$ | $\frac{1.5}{4}\frac{\kappa_3}{\kappa_{end}} Y$ | $\frac{1.5}{2}\frac{\kappa_{20}}{\kappa_{end}}\frac{\lambda}{\gamma}$ |

$\xi \approx 1.5$ and $\gamma T \approx 4$ (with $\gamma = 365/7$, $y^{-1}$ and $T = 1/12$, $y^{-1}$). For municipalities with data, quantities are computed at the municipal-monthly level. In the absence of data for a municipality, provincial and/or yearly estimates are used as a baseline to avoid classifying municipalities without data as no-risk areas. Reported cases refer to detected viremic imported cases.

### Spatio-temporal risk quantification

To further analyze the data, we calculate the maximum likelihood estimator $\hat{\lambda}$ (as given in Eq (7)), which provides an estimate of the expected rate of imported cases. On average, $\hat{\lambda} \approx 0.04$, leading to an expected number of imported cases, $\langle Y \rangle \approx 0.3$ (using Eq (9)). Even in Bizkaia, where the highest value recorded in 2023 was $\hat{\lambda} \approx 0.07$, $\langle Y \rangle$ still remains low at approximately 0.5. This aligns with our assumption of a low number of expected infected cases, consistent with the non-endemic hypothesis.

This mean value of 0.5 autochthonous cases implies that, statistically, one locally transmitted case is expected every two years, assuming conditions remain unchanged. However, due to the stochastic nature of disease transmission, an actual case may occur sooner or later than the average suggests. While 0.5 cases may seem low and does not indicate an immediate risk, it highlights the potential for local outbreaks in the Basque Country, emphasizing the need for ongoing monitoring and preparedness. The estimated $\hat{\lambda}$ values for each province of the Basque Country are presented in Table 3.

Regarding the vector abundance, Table 4 reports the largest egg counts per ovitrap, to have an understanding of the ongoing invasion and establishment of *Aedes* species. In 2019 and 2022, the highest egg counts occurred during the warm season –July, August, and September– as anticipated, whereas in 2023 were registered in September and October, which may be connected to environmental changes. As observed, there is a general increase in the presence of the vector across all provinces of the Basque Country over the years. It first entered through Gipuzkoa, gradually expanded into Bizkaia, and finally reached Araba, where the highest egg counts are now comparable to those recorded in Bizkaia in 2019, showing a signature of the ongoing expansion of the vector across the Basque territory.

Fig 4 shows the evolution of risk over time, highlighting the increasing threat of arbovirus diseases across all provinces, particularly in urban and peri-urban areas of the Basque region.

In 2019, the situation in Gipuzkoa showed little variation throughout the months, with a province-wide risk level comparable to that of Bizkaia, while data for Araba remained insufficient. Only Irun, a municipality in Gipuzkoa, exhibited a slight but notable increase that year. By 2022, the risk had continued to rise, with Gipuzkoa leading, followed by Bizkaia and Araba. As in previous years, the highest values were recorded in the border municipalities of

**Table 3. Estimates of the maximum likelihood estimator for the effective constant rate of imported cases, $\hat{\lambda}$ (in units of day$^{-1}$).**

| Year | Bizkaia | Gipuzkoa | Araba |
|------|---------|----------|-------|
| 2019 | 0.0466 | 0.0384 | - |
| 2022 | 0.0575 | 0.0493 | 0.0274 |
| 2023 | 0.0741 | 0.0549 | 0.0439 |

Due to only one imported case being reported in Araba for 2019, there was insufficient data to derive statistics related to inter-event times.

**Table 4. The three highest positive mosquito egg counts per oviposition stick (collected over a 15-day period) for each Basque province.**

| Year | Bizkaia | Gipuzkoa | Araba |
|------|---------|----------|-------|
| 2019 | 134, 105, 98 | 260, 258, 240 | - |
| 2022 | 1005, 382, 277 | 1000, 596, 594 | 17, 2 |
| 2023 | 1005, 958, 913 | 1216, 1100, 1000 | 142, 129, 117 |

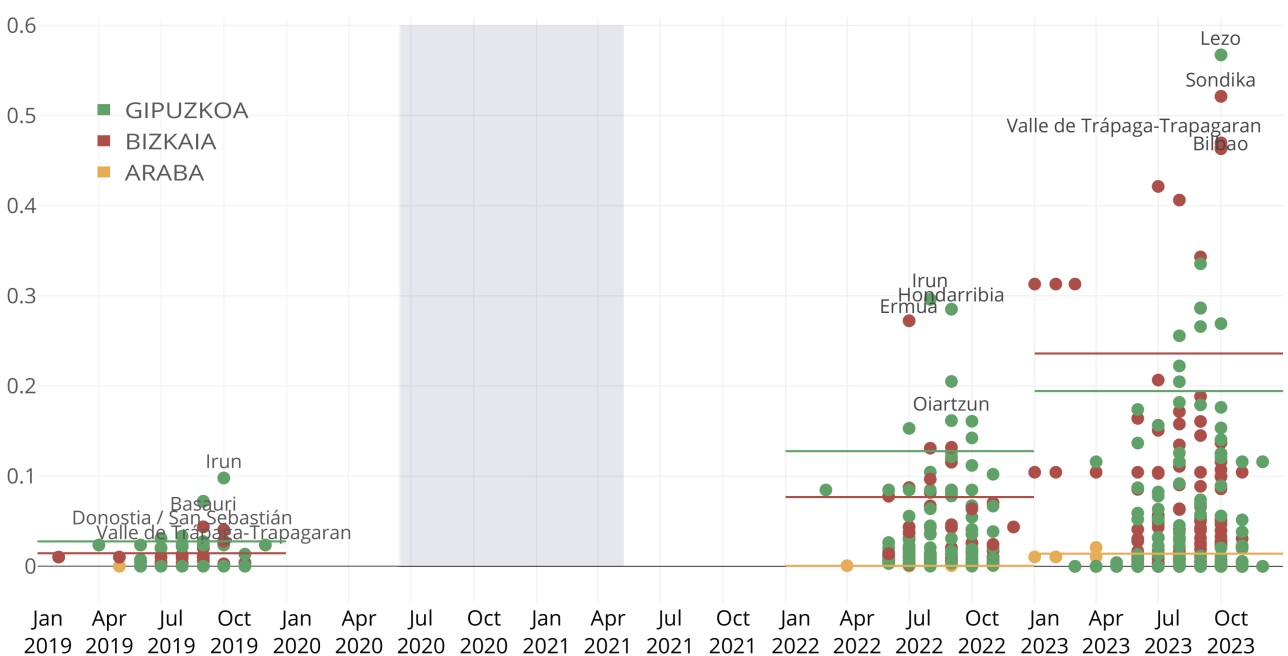

**Fig 4. Risk estimation of the expected number of autochthonous cases at the municipal level, based on the SIRUVY model.** Risk is computed using the formulas from Table 2. Only points that deviate from the provincial estimate are shown, with horizontal lines representing the annual baseline level for each province. Colors indicate the corresponding province, and the top four highest-risk municipalities each year are labeled. The period highlighted in light purple corresponds to the COVID-19 lockdown period.

Gipuzkoa, particularly Irun and Hondarribia. However, Ermua in Bizkaia exhibited comparable risk levels, likely due to its proximity to Eibar in Gipuzkoa. Both municipalities are part of the same urban area within the Deba river basin, commonly grouped together as the Bajo Deba region.

By 2023, Bilbao (the capital of Bizkaia) and nearby municipalities like Sondika and Trapagaran surpassed historical peaks previously seen in Gipuzkoa, with the highest values now reported in 2023 (excluding Lezo). This marks not only an increase in risk but also a shift in potential hotspots. The risk levels in Gipuzkoa and Araba also continued to climb, surpassing their historical records, signaling a worsening trend.

Climate change, together with other factors, is altering environmental conditions and enabling mosquitoes to adapt and survive in regions that were previously unsuitable. While disease outbreaks have not yet been confirmed and the timing of potential future outbreaks remains uncertain, these changing conditions increasingly favor mosquito population growth, thereby elevating the risk of outbreaks. This risk is particularly concerning in areas that may be unprepared to face such emerging threats and the associated public health challenges.

Given their effectiveness in visualizing geospatial data and identifying spatial patterns, choropleth maps are valuable tools for communicating results to non-technical managers and policymakers. In Fig 5, we map the spatial distribution of key risk factors, including the relative abundance of mosquitoes ($k$), the expected number of imported infected cases ($\langle Y \rangle$), and the risk of autochthonous transmission for a month with high activity, as described in the section "Risk Evaluation for the Chosen Spatio-Temporal Scales" above. For the chosen month, the blue-scale map shows that the highest mosquito populations are in Gipuzkoa, particularly Hondarribia and Irun, located on the border with France. The violet-scale map

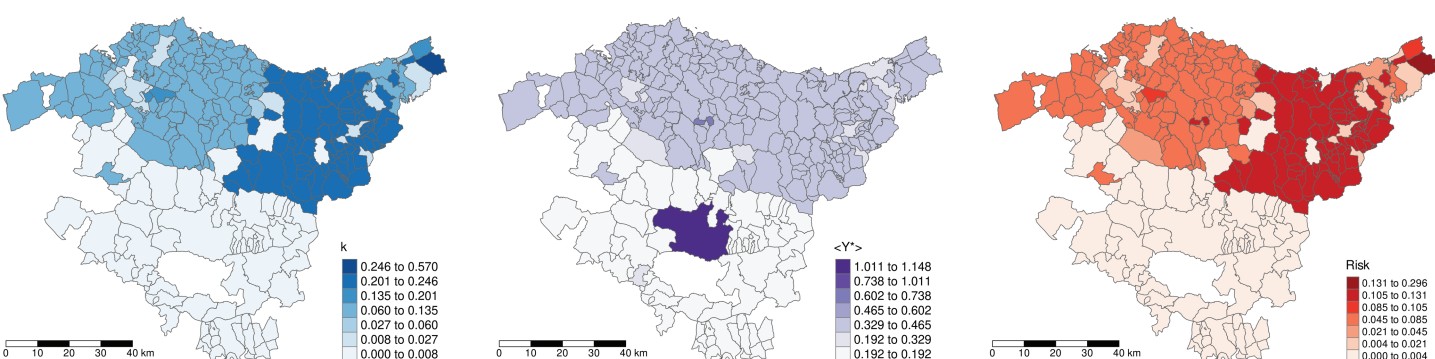

**Fig 5. Maps for August 2022 showing relative mosquito abundance *k* (blue), expected number of imported viremic cases $\langle Y \rangle$ (purple), and the resulting risk map (red).** Data are aggregated at the municipal and monthly level; where data is missing, provincial and yearly averages are used to fill gaps. Monthly maps of relative *Aedes* mosquito abundance and expected imported viremic cases at the municipal level in the Basque Country for 2019, 2022, and 2023 are available in S2 Fig and S3 Fig. Base map layer from the Basque Government (Eusko Jaurlaritza / Gobierno Vasco) resource https://www.euskadi.eus/limites-administrativos-del-pais-vasco/web01-ejeduki /es/, under CC BY 4.0 https://creativecommons.org/licenses/by/4.0/.

reveals that imported cases have been randomly reported across various localities in all provinces. For this particular month, 5 cases in Vitoria, 3 in Durango, 2 in Eibar, and 8 in other regions. The red-scale map provides details of the quantified risk assessment. The maps for the remaining months and years are available in S2 Fig, S3 Fig, and S4 Fig.

Fig 6 illustrates the spatial progression of mosquito-borne infections during the warm seasons of 2019, 2022, and 2023. Sparse entomological monitoring during cold seasons limits the scope of the analysis regarding annual seasonality. Preliminary data confirm that no *Aedes* activity (eggs in ovitraps) has been detected between December 2023 and May 2024.

These maps illustrate the increase and temporal shifts in risk levels while effectively conveying insights into mosquito-borne disease risk. To ensure comparability and clarity in visual interpretation, a consistent color scale with uneven breaks has been applied across all maps of each type shown in Fig 5, though alternative classification methods [53] could enhance differentiation between classes for a more detailed analysis. When interpreting these maps, it is important to account for map area bias, which may overemphasize larger regions while minimizing smaller ones. Nonetheless, despite the data being aggregated by municipalities, the risk calculation does not depend on municipal area size.

## Scenario-based risk maps: Spanish classification and alternative approaches

Risk maps are invaluable in public health for analyzing and addressing the spatio-temporal dynamics of epidemiological threats. By tracking disease spread and identifying high-risk areas, these maps help scientists and health authorities translate complex epidemiological data into clear, actionable formats for decision-makers and the public [54]. To assess scenarios, some risk maps use qualitative categorizations to highlight key attributes and prioritize interventions across regions. Their flexibility allows for the integration of diverse data sources and adaptation to emerging scenarios or evolving evaluation criteria, supporting evidence-based decision-making and strengthening public health strategies.

This capability is critical for guiding surveillance efforts, allocating resources efficiently, and designing targeted interventions that mitigate the impact of infectious diseases and other public health threats.

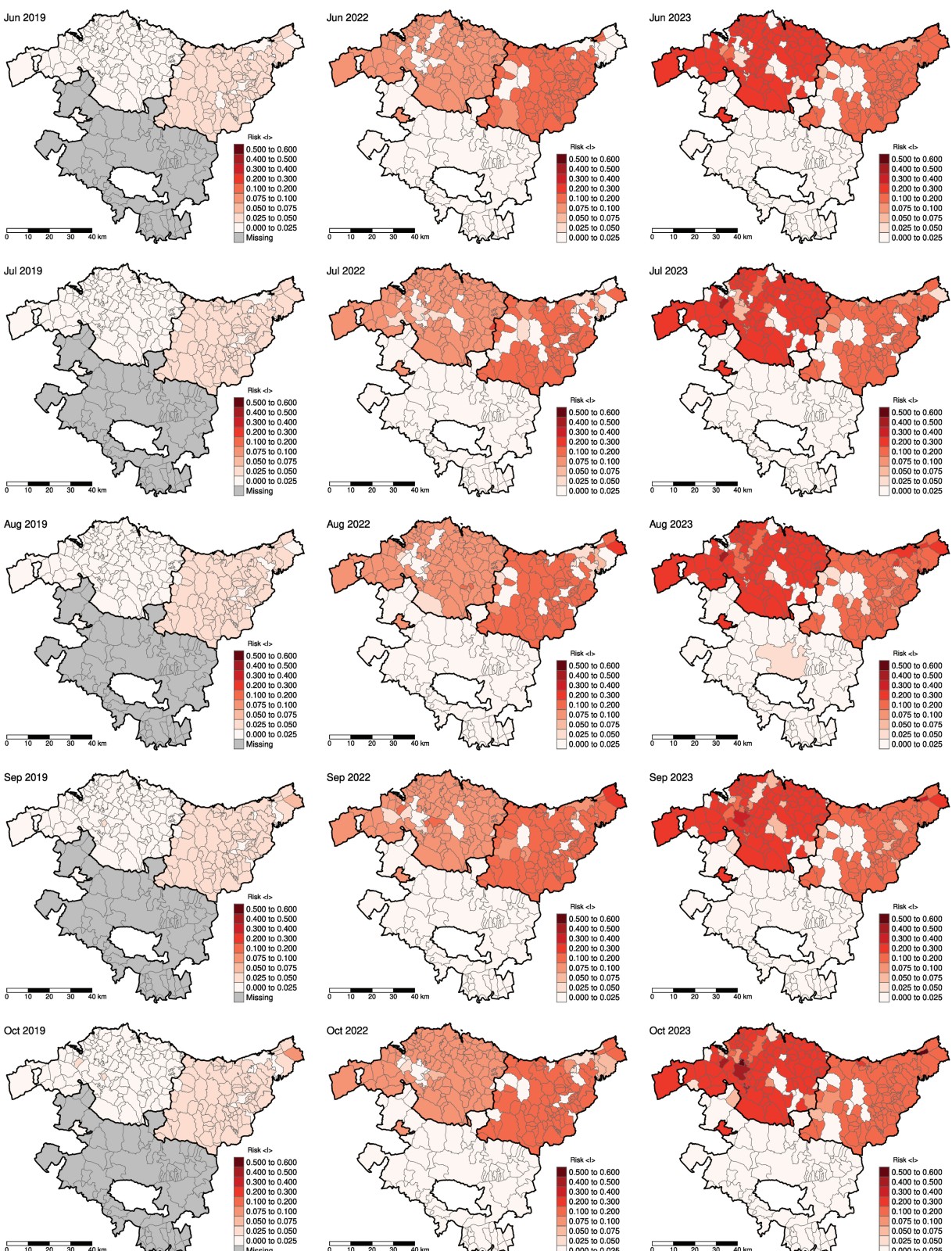

**Fig 6. Risk maps at the municipal level for the warm seasons of 2019, 2022, and 2023.** A consistent color scale is applied across all maps to facilitate comparison. Base map layer from the Basque Government (Eusko Jaurlaritza / Gobierno Vasco) resource https://www.euskadi.eus/limites-administrativos-del-pais-vasco/web01-ejeduki/es/, under CC BY 4.0 https://creativecommons.org/licenses/by/4.0/.

It is important to highlight that scenario-based maps, which rely on categorical variables linked to explanatory factors, provide a qualitative representation of risk rather than a precise quantitative measure. This approach differs from the risk map discussed in the previous section, which quantifies risk using a continuous variable to estimate the expected number of autochthonous cases for a given location and time period. However, both mapping strategies are valuable and can be used effectively depending on the analysis's objectives.

In the context of defining public health objectives and strategies to control diseases transmitted by *Aedes albopictus*, the Spanish Ministry of Health, through its National Plan for the Prevention, Surveillance, and Control of Vector-Borne Diseases [55], established risk categories based on criteria outlined in Table 5. These categories consider the presence of the invasive mosquito species, confirmation of imported arboviral cases, and/or the occurrence of autochthonous infected cases.

In Fig 7, we present the scenario-based map derived from Table 5 for 2022 and 2023. While these maps effectively highlight key areas near the capitals of Bizkaia and Gipuzkoa and along the French border, they lack the detail needed to prioritize intervention measures. As demonstrated in Figs 4 and 6, potential outbreak hotspots can shift significantly from one year to the next. This emphasizes the importance of using different risk assessment approaches, with model-based estimations providing strong support for more targeted decision-making.

In some cases, categorical variables require expert consensus to avoid ambiguity. For example, categories such as established *Aedes albopictus* or active *Aedes albopictus* can be interpreted in various ways. In this study, we define *Aedes albopictus* as "established" based on criteria used by the entomological study team at NEIKER. Specifically, a municipality is considered established if positive ovitrap counts are recorded in at least two distinct locations within the municipality for two consecutive years. Additionally, if a municipality is adjacent to areas where the mosquito is already established, it may be classified as established if a significant proportion of ovitraps within the municipality yield positive results.

Drawing from the categories outlined in the Spanish National Plan (see Table 5), we propose alternative scenario-based maps (see Fig 8) using three different classification categories, as detailed in the tables of Fig 8. These categories are based on the presence or absence of infected cases and mosquito eggs. *Alternative A* relies solely on data from the specific month. In contrast, *Alternative B* incorporates historical records of *Aedes* egg counts, denoted by the

**Table 5. Entomological risk classification for diseases transmitted by *Aedes albopictus*, as proposed by the Spanish Ministry of Health [55].**

| 0 | ***Aedes albopictus* not identified.** |
|---|---|
| 0a | Periodic entomological surveillance is carried out in optimal areas for the presence of the species, and its presence has not been confirmed. |
| 0b | No entomological surveillance is carried out, and there is no previous data on the presence of the species in the area of interest. |
| 0c | There are municipalities bordering the area of interest that have established populations of the species. |
| 1 | **Recent and isolated detection of *Aedes albopictus*.** The species is not yet considered established in that area. |
| 2 | **Established *Aedes albopictus*.** |
| 2a | No autochthonous cases have been detected. Imported cases may be detected, and recommendations will be established based on the viremic status of the cases. |
| 2b | Detection of an autochthonous case of a disease transmitted by this vector or one or several case clusters. |
| 2c | Epidemic transmission in an area. Wide distribution of human cases not linked to clusters, without geographic or temporal link between them. |

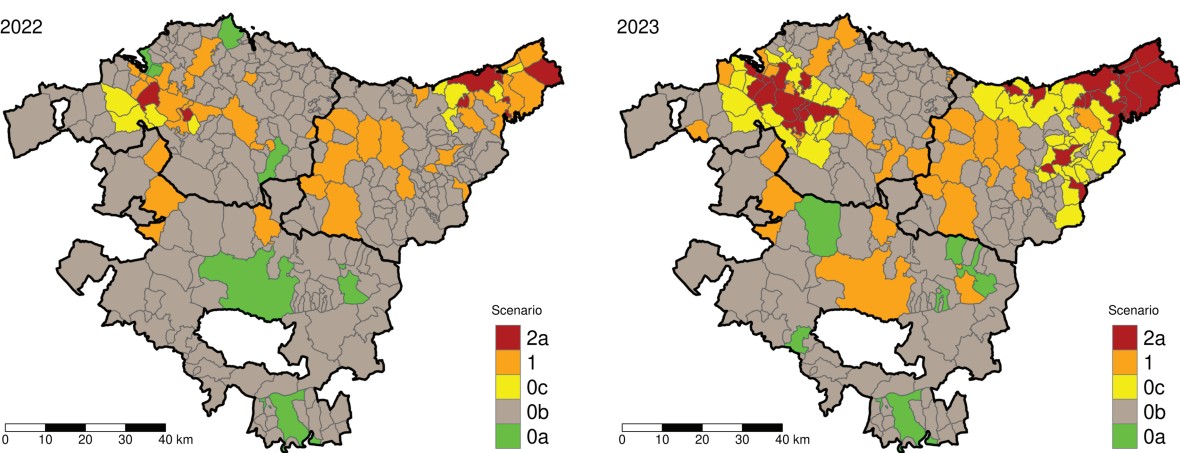

**Fig 7. Scenario-based risk maps for 2022 and 2023, based on the entomological classification proposed by the Spanish Ministry of Public Health (Table 5), considering *Aedes* species in general, including *Aedes albopictus* and *Aedes japonicus*.** The map for 2019 is available in the S5 Fig. Categories 2b and 2c were excluded as they do not apply to a non-endemic setting without autochthonous cases. Base map layer from the Basque Government (Eusko Jaurlaritza / Gobierno Vasco) resource https://www.euskadi.eus/limites-administrativos-del-pais-vasco/web01-ejeduki/es/, under CC BY 4.0 https://creativecommons.org/licenses/by/4.0/.

*(hist)* label. *Alternative C* combines both historical egg count data and reported cases. While *Alternative A* is useful for detecting temporal variations by comparing maps from different months, *Alternatives B and C* are better suited for assessing long-term trends or evaluating the impact of specific public health measures. In Fig 8, the scenario-based maps at the municipal level for each of these three alternatives are shown.

The scenario-based maps generated with these alternative categories offer more detailed insights into the evolution of risk. They provide additional information that may not be captured by the Public Health Department's classification, offering a broader perspective.

## Discussion

Non-endemic scenarios show distinct epidemiological features compared to endemic regions. In endemic areas, imported cases have limited impact, and modeling requires explicit strain-structured compartments to capture complex local viral co-circulation. Conversely, non-endemic regions lack sustained local transmission, with only sporadic imported and autochthonous cases, and populations remain largely immunologically naive. Mosquito vectors may biologically co-transmit multiple viruses but often carry only one strain [56], simplifying the dynamics. This makes the SIRUVY modeling framework particularly suitable, yielding a straightforward risk expression where the expected number of local cases, $\langle I \rangle$, indicates outbreak potential.

Challenges in non-endemic areas include limited long-term surveillance and data quality, as these regions often prioritize more immediate health concerns. Data inconsistencies hinder early detection and response, underscoring the value of insights drawn from endemic settings while accounting for local context differences.

Our risk estimator relies solely on mosquito abundance and imported case counts, making it especially useful where detailed covariate data are scarce or uncertain. By implicitly capturing environmental and ecological factors through empirical records, the model balances simplicity with realism, enhancing robustness and interpretability.

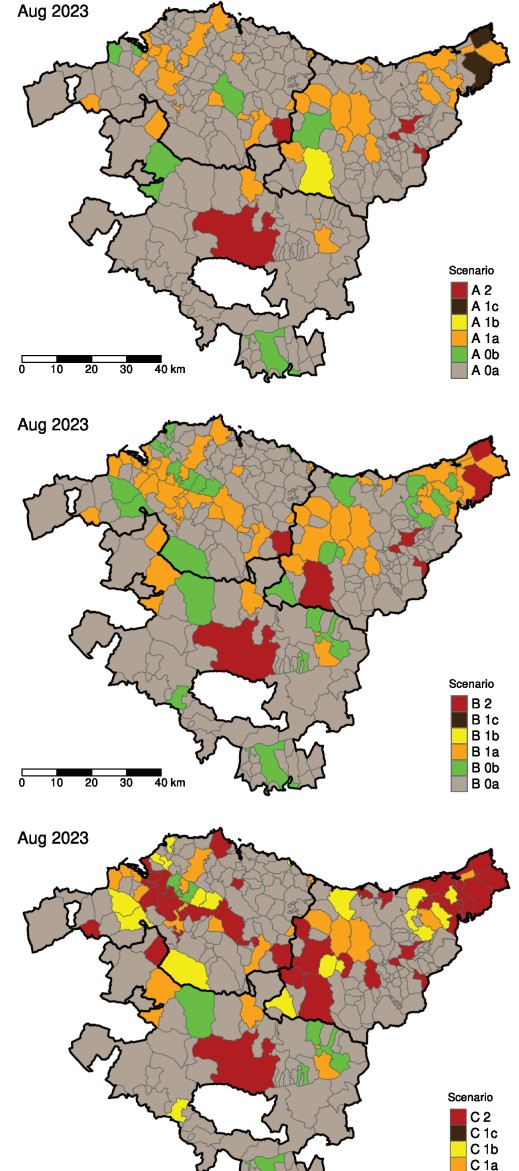

| A. Cases (month) - Aedes (month) | |
|---|---|
| Label | Description |
| 2 | Cases Yes, Aedes presence |
| 1c | Cases Yes, No Aedes surveillance |
| 1b | Cases Yes, No Aedes presence |
| 1a | Cases No, Aedes presence |
| 0b | Cases No, No Aedes presence |
| 0a | Cases No, No Aedes surveillance |

| B. Cases (month) - Aedes (hist) | |
|---|---|
| Label | Description |
| 2 | Cases Yes, Aedes presence (hist) |
| 1c | Cases Yes, No Aedes surveillance |
| 1b | Cases Yes, No Aedes presence (hist) |
| 1a | Cases No, Aedes presence (hist) |
| 0b | Cases No, No Aedes presence (hist) |
| 0a | Cases No, No vector surveillance |

| C. Cases (hist) - Aedes (hist) | |
|---|---|
| Label | Description |
| 2 | Cases Yes (hist), Aedes presence (hist) |
| 1c | Cases Yes (hist), No Aedes surveillance |
| 1b | Cases Yes (hist), No Aedes presence (hist) |
| 1a | Cases No (hist), Aedes presence (hist) |
| 0b | Cases No (hist), No Aedes presence (hist) |
| 0a | Cases No (hist), No Aedes surveillance |

**Fig 8. Alternative scenario-based maps for each proposed classification.** Monthly scenario A-, B-, and C-based Aedes-borne disease risk maps in the Basque Country, based on the entomological classification proposed by the Spanish Ministry of Public Health for August 2023. Maps for the remaining months and years are available in S6 Fig, S7 Fig, and S8 Fig. Base map layer from the Basque Government (Eusko Jaurlaritza / Gobierno Vasco) resource https://www.euskadi.eus/limites-administrativos-del-pais-vasco/web01-ejeduki/es/, under CC BY 4.0 https://creativecommons.org/licenses/by/4.0/.

We aligned analysis to the operational scale of local health authorities using monthly municipal data. When such data were unavailable, broader spatial or temporal aggregations were used without interpolation to preserve data integrity. Future models could incorporate seasonality or other covariates if long-term, high-resolution data become available. In our study area, seasonal modeling was less relevant due to the absence of mosquitoes in colder months.

Ovitrap egg counts served as a practical proxy for adult mosquito abundance, supported by literature linking egg numbers to biting female populations. This method is well suited for low-density, non-endemic settings, providing a cost-effective tool for early warning and targeted control.

Underreporting of cases likely leads to underestimation of importations, but incorporating temporal case distributions improves risk estimates by capturing stochastic dynamics more accurately. We modeled importations as a homogeneous Poisson process per month - a justifiable simplification given sparse data and lack of strong evidence for seasonality or alternative patterns.

Importation signals aggregate diverse sources and hemispheres, smoothing out clear seasonal trends. Asymptomatic and missed infections further complicate detection, supporting the conservative Poisson assumption.

The risk estimator is an average measure with no formal upper bound - $\langle I \rangle^* \in [0, +\infty)$ - which complicates the definition of thresholds for classifying risk levels, as in other approaches using loosely defined empirical cutoffs. It does not impose fixed spatial or temporal scales, and its accuracy depends heavily on the resolution and alignment of entomological and epidemiological data, which are often mismatched in practice. Despite these limitations, the estimator offers a practical comparative framework to identify relative changes and spatial patterns, supporting the prioritization of surveillance and intervention rather than providing precise forecasts.

As a comparative and data-informed framework, our tool is intended to support the prioritization of vector surveillance and early intervention, rather than to provide precise forecasts of outbreak magnitude. While any risk assessment system may be sensitive to uncertainties in input data or assumptions, our model emphasizes relative changes and spatial trends rather than absolute case counts. This design helps reduce the risk of misinterpretation and supports its use as a complementary input for public health decision-making, especially in non-endemic settings where timely and targeted responses are critical.

Continuous entomological surveillance is essential in regions undergoing environmental or urban change, as exemplified by the Basque Country's monitoring efforts since the detection of *Aedes albopictus* in 2013. The adaptability of our framework can be further enhanced by integrating environmental covariates affecting mosquito suitability [30,38,51,57–62], human mobility patterns and traveler influx [9,63–65], as well as advanced spatiotemporal modeling techniques. Hybrid and multimodel approaches, supported by increasingly harmonized multisource datasets [66], would improve the reliability of risk assessments, strengthen surveillance, and refine scenario forecasting, further enhancing public health planning within the conceptual framework developed here.

## Conclusion

We proposed a spatio-temporal quantifier of the expected number of autochthonous cases, based on the deterministic solution of the SIRUVY model, as a proxy for assessing the risk of arboviral diseases transmitted by *Aedes* mosquitoes in non-endemic regions. Applied to the Basque Country and tailored to the available data, our framework accommodates different spatial (municipal, provincial) and temporal (monthly, annual) scales, producing time series and choropleth maps to guide targeted public health interventions.

Our findings, based on entomological and epidemiological data up to 2023, reveal an upward trend in both mosquito abundance and imported viremic cases. Risk estimates highlight Irun and neighboring municipalities as persistent hotspots since 2013, with recent increases in vector presence shifting the highest risk levels to Bizkaia. Despite a doubling of

risk levels from 2022 to 2023, no local cases have been reported to date, and mosquito activity remains absent in colder months, reaffirming the Basque Country's non-endemic status. These observed trends highlight the limitations of linear forecasting methods and underscore the importance of continuous, data-driven risk assessments to capture the evolving patterns of vector spread and associated disease risks.

The proposed risk estimator is simple to compute, interpretable, and adaptable to data-limited settings, making it a valuable tool for early-stage surveillance and public health planning. Unlike more complex models, it relies on empirical data without requiring unknown parameters, offering a robust alternative for non-endemic areas facing emerging disease threats.

Broader application of this framework requires standardized, high-resolution entomological and epidemiological data, which are often lacking in many regions. Beyond its methodological value, this work emphasizes the need to strengthen surveillance systems. Improved data availability will not only enhance local risk assessments but also support cross-regional analyses and timely public health responses amid increasing environmental and urbanization pressures.

## Supporting information

**S1 Fig. Monthly maps of reported cases and highest *Aedes* mosquito egg counts at the municipal level in the Basque Country for 2019, 2022, and 2023.**
(PDF)

**S2 Fig. Monthly maps of relative *Aedes* mosquito abundance at the municipal level in the Basque Country for the years 2019, 2022, and 2023.**
(PDF)

**S3 Fig. Monthly maps of the expected number of imported viremic cases at the municipal level in the Basque Country for 2019, 2022, and 2023.**
(PDF)

**S4 Fig. Monthly estimated *Aedes*-borne disease risk maps at the municipal level in the Basque Country for 2019, 2022, and 2023.**
(PDF)

**S5 Fig. Scenario-based *Aedes*-borne disease risk maps in the Basque Country, by municipality, for 2019, 2022, and 2023, based on the entomological classification proposed by the Spanish Ministry of Public Health.**
(PDF)

**S6 Fig. Monthly scenario *A-based Aedes*-borne disease risk maps in the Basque Country, based on the entomological classification proposed by the Spanish Ministry of Public Health.**
(PDF)

**S7 Fig. Monthly *scenario B*-based *Aedes*-borne disease risk maps in the Basque Country, based on the entomological classification proposed by the Spanish Ministry of Public Health.**
(PDF)

**S8 Fig. Monthly scenario *C-based Aedes*-borne disease risk maps in the Basque Country, based on the entomological classification proposed by the Spanish Ministry of Public Health.**
(PDF)

## Acknowledgments

We wish to extend our acknowledgments to Oscar Goñi Laguardia from Dirección de Salud Pública for his fruitful discussions, and to Madalen Oribe Amores, Unidad de Vigilancia Epidemiológica de Bizkaia, for her cooperation in providing the requested epidemiological data that were essential for carrying out this research. We also thank Patricia Vázquez for her laboratory work counting *Aedes* eggs, and Ana L. García-Pérez for her efforts in coordinating and leading tiger mosquito surveillance until 2022.

## Author contributions

**Conceptualization:** Bruno V. Guerrero, Vanessa Steindorf, Nico Stollenwerk, Maira Aguiar.

**Data curation:** Bruno V. Guerrero, Vanessa Steindorf, Rubén Blasco-Aguado, Luís Mateus, Aitor Cevidanes, Jesús F. Barandika, Ana Ramíirez de La Peciña Perez, Joseba Bidaurrazaga Van-Dierdonck, Jesús Angel Ocio Armentia.

**Formal analysis:** Bruno V. Guerrero, Vanessa Steindorf, Rubén Blasco-Aguado, Luís Mateus, Aitor Cevidanes, Jesús F. Barandika, Nico Stollenwerk, Maira Aguiar.

**Funding acquisition:** Maira Aguiar.

**Investigation:** Bruno V. Guerrero, Vanessa Steindorf, Nico Stollenwerk, Maira Aguiar.

**Methodology:** Bruno V. Guerrero, Vanessa Steindorf, Rubén Blasco-Aguado, Luís Mateus, Aitor Cevidanes, Jesús F. Barandika, Nico Stollenwerk, Maira Aguiar.

**Project administration:** Maira Aguiar.

**Supervision:** Maira Aguiar.

**Validation:** Maira Aguiar.

**Visualization:** Bruno V. Guerrero, Nico Stollenwerk.

**Writing – original draft:** Bruno V. Guerrero.

**Writing – review & editing:** Bruno V. Guerrero, Vanessa Steindorf, Rubén Blasco-Aguado, Luís Mateus, Aitor Cevidanes, Jesús F. Barandika, Ana Ramíirez de La Peciña Perez, Joseba Bidaurrazaga Van-Dierdonck, Jesús Angel Ocio Armentia, Nico Stollenwerk, Maira Aguiar.

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
