## [Decision Letter · Decision Letter 0]

20 Apr 2025

PNTD-D-25-00343

Assessing the Spatio-Temporal Risk of Aedes-Borne Arboviral Diseases in Non-Endemic Regions: The Case of Northern Spain

Dear Dr. Aguiar,

Thank you for submitting your manuscript to PLOS Neglected Tropical Diseases. After careful consideration, we feel that it has merit but does not fully meet PLOS Neglected Tropical Diseases's publication criteria as it currently stands. Therefore, we invite you to submit a revised version of the manuscript that addresses the points raised during the review process.

Please submit your revised manuscript within 60 days Jun 19 2025 11:59PM. If you will need more time than this to complete your revisions, please reply to this message or contact the journal office at plosntds@plos.org. Please include the following items when submitting your revised manuscript:

We look forward to receiving your revised manuscript.

Kind regards,

Ran Wang, M.D.

Academic Editor

Victoria Brookes

Section Editor

Shaden Kamhawi

co-Editor-in-Chief

Paul Brindley

co-Editor-in-Chief

**Journal Requirements:**

At this stage, the following Authors/Authors require contributions: Bruno V. Guerrero, Vanessa Steindorf, Rubén Blasco-Aguado, Luis Mateus, Aitor Cevidanes, Jesus F. Barandika, Ana Ramiırez de La Pecina Perez, Joseba Bidaurrazaga Van-Dierdonck, Jesus Angel Ocio Armentia, Nico Stollenwerk, and Maira Aguiar. Please ensure that the full contributions of each author are acknowledged in the "Add/Edit/Remove Authors" section of our submission form.

5) We have noticed that you have uploaded Supporting Information files, but you have not included a list of legends. Please add a full list of legends for your Supporting Information files after the references list.

Potential Copyright Issues:

i) Figure 2. Please confirm whether you drew the images / clip-art within the figure panels by hand. If you did not draw the images, please provide (a) a link to the source of the images or icons and their license / terms of use; or (b) written permission from the copyright holder to publish the images or icons under our CC BY 4.0 license. Alternatively, you may replace the images with open source alternatives. See these open source resources you may use to replace images / clip-art:

ii) Figures 1C, 1D, (5-8), and the Figures in the Supplementary Material file. Please (a) provide a direct link to the base layer of the map (i.e., the country or region border shape) and ensure this is also included in the figure legend; and (b) provide a link to the terms of use / license information for the base layer image or shapefile. We cannot publish proprietary or copyrighted maps (e.g. Google Maps, Mapquest) and the terms of use for your map base layer must be compatible with our CC BY 4.0 license.

7) We note that you have indicated that there are restrictions to data sharing for this study. PLOS only allows data to be available upon request if there are legal or ethical restrictions on sharing data publicly. For more information on unacceptable data access restrictions, please see https://journals.plos.org/plosntds/s/data-availability#loc-unacceptable-data-access-restrictions.

b) If there are no restrictions, please upload the minimal anonymized data set necessary to replicate your study findings to a stable, public repository and provide us with the relevant URLs, DOIs, or accession numbers. For a list of recommended repositories, please see https://journals.plos.org/plosone/s/recommended-repositories. You also have the option of uploading the data as Supporting Information files, but we would recommend depositing data directly to a data repository if possible.

8) Please amend your detailed Financial Disclosure statement. This is published with the article. It must therefore be completed in full sentences and contain the exact wording you wish to be published.

9) Please ensure that the funders and grant numbers match between the Financial Disclosure field and the Funding Information tab in your submission form. Note that the funders must be provided in the same order in both places as well. Currently, the order of the funders is different in both places. In addition, "Department of Food, Rural Development, Agriculture and Fisheries, and the Departament of Health of the Basque Government, the Ministry of Health, Social Policy, and Equality of the Government of Spain" are missing from the Funding Information tab.

**Comments to the Authors:**

**Please note that one of the reviews is uploaded as an attachment.**

**Reviewers' Comments:**

Reviewer's Responses to Questions

**Key Review Criteria Required for Acceptance?**

**Methods**

-Are the objectives of the study clearly articulated with a clear testable hypothesis stated?

-Is the study design appropriate to address the stated objectives?

-Is the population clearly described and appropriate for the hypothesis being tested?

-Is the sample size sufficient to ensure adequate power to address the hypothesis being tested?

-Were correct statistical analysis used to support conclusions?

-Are there concerns about ethical or regulatory requirements being met?

Reviewer #1: Regarding the study data, it is mentioned: "Due to ethical considerations and commercial sensitivity, these data are not publicly available." Given this restriction, is there an Institutional Review Board (IRB) approval or an equivalent ethics committee approval for the use of this data? Clarifying this would help ensure compliance with ethical research standards and data protection guidelines. Additionally, while the data may not be publicly available, it is important to provide a clearer understanding of its nature, scope, and any preprocessing steps taken. This would enhance transparency and allow readers to better assess the study's methodology and findings.

To enhance the clarity and impact of the study, it is advisable to explicitly state the hypothesis and objectives.

It is important to provide general details about the analyzed population, as this contextualizes the study and ensures a better understanding of its scope and applicability.

In lines 9-10 when you mention “Our refined SIRUV human-vector compartmental model incorporates stochastic formulations to capture fluctuations in imported cases, mosquito density, and autochthonous transmission risk - critical factors in outbreak emergence” you could include the SIRUVY model as this “Our refined SIRUV human-vector compartmental model (SIRUVY) incorporates stochastic formulations to capture fluctuations in imported cases, mosquito density, and autochthonous transmission risk - critical factors in outbreak emergence” I believe it is important to explicitly state "SIRUVY" in this way, as it highlights the novelty of the model.

The sentence in lines 26 through 28 “Climate change, combined with rising international human mobility and trade, not only facilitates the proliferation of disease vectors but also drives the introduction of pathogens into naïve areas, increasing interactions between infected individuals and susceptible populations” could be misinterpreted to suggest that vector-borne diseases (VBDs) are directly transmitted between infected individuals and susceptible populations. To clarify that the transmission occurs through vectors, you might consider rewording it like this: "Climate change, combined with rising international human mobility and trade, not only facilitates the proliferation of disease vectors, but also drives the introduction of pathogens into naïve areas, increasing the risk of local vector-mediated transmission to susceptible populations."

In line 33 you could change “arrival of imported cases” for “introduction of imported cases”. If you choose to implement this suggestion and decide on a specific term, I recommend using the same term consistently throughout the article, provided this comment is applicable.

In line 34 “autochthonous cases—locally transmitted infections” sounds redundant. You could use “locally transmitted infections” instead.

On lines 75-76, I would use the word adaptively “Once the relevant data are available, this framework can be adaptively applied to other regions experiencing the introduction of Aedes-borne diseases.”

On lines 96–97, the description of the database is limited. To enhance clarity and reproducibility, consider including additional details such as the total number of reported cases, the period covered by the dataset, and key variables recorded. Additionally, specify whether the cases are confirmed, suspected, or both. If the dataset is publicly available, please indicate whether it is open-source and provide a reference if possible. Similarly, for the entomological data mentioned on lines 97–98, please clarify the nature of the data provided by NEIKER.

On line 106, would it be useful to specify the epidemiological weeks of the years mentioned?

In lines 125-133 when you mention “Additionally, there is a steady external import of new infected cases Y, introduced at a constant arrival rate ◯ϱ.” wouldn’t this be deterministic since new cases are introduced at a constant rate? If there is any stochastic variability in the importation process, consider clarifying whether ◯ϱ represents an expected average rate at a certain point in time, rather than a strictly fixed value. My understanding is that S0, U0 and Y0, serve as initial conditions that shape the model's behavior. However, when applied in different contexts, they can act as adaptive variables rather than fixed parameters, allowing the model to be used in a contextual rather than a purely replicative way. It is key to emphasize that Y0, is crucial for non-endemic regions where outbreaks are seeded by travelers. The contextual number of imported cases will vary over time depending on travel patterns, seasonality, and other external factors. Based on this, these adaptive variables enable the model to be applied in a contextual, time-based, and adaptive manner across different settings, rather than simply replicating it unchanged. It is essential to highlight this point in the article as your audience may not be familiar with this. This would allow to overstress that, as stated in line 63, “deterministic and stochastic models are not designed as direct risk estimators” and as stated in lines 158-159 “However, while this result is derived from a deterministic model, stochasticity is essential to account for the irregular behavior and large fluctuations observed in real disease incidence data”

In lines 173-175 you state “This variability emphasizes the importance of considering stochastic effects in low-endemicity settings, where random fluctuations can greatly determine observed outbreak dynamics and hinder accurate predictions in real-world scenarios.” However, in a dynamic system determination would imply that there is an absolute behavior under certain contextual elements. In that regards you could say “This variability emphasizes the importance of considering stochastic effects in low-endemicity settings, where random fluctuations can greatly influence the determinable nature of observed outbreak dynamics and hinder accurate predictions in real-world scenarios."

Reviewer #2: Yes

Reviewer #3: (No Response)

**Results**

-Does the analysis presented match the analysis plan?

-Are the results clearly and completely presented?

-Are the figures (Tables, Images) of sufficient quality for clarity?

Reviewer #1: In figure 1, the words “imported cases” are overlapping with the maps. You could use some coding to avoid that overlapping.

In line 274: "In 2019 and 2022, the highest egg counts occurred during the warm season—July, August, and September—as anticipated." hot season might be considered too colloquial. A more precise and appropriate term would be "warm season", "summer months", or "peak temperature months".

Regarding Figure 5 and the preceding lines, it is important to consider the broader perspective: climate change is driving environmental shifts that enable mosquitoes to adapt to regions that were previously unsuitable for their survival. While this process is unfolding, and it may be difficult to definitively state that dengue outbreaks are occurring, the changing conditions are increasingly favorable for mosquito populations to thrive. This, in turn, heightens the likelihood of future outbreaks. Although the exact timing of such outbreaks remains unpredictable, the prevailing conditions strongly suggest that their occurrence is becoming more probable. This has significant public health implications, as expanding mosquito habitats may lead to an increased burden of vector-borne diseases in regions that lack prior experience in managing them. Strengthening surveillance, preparedness, and targeted interventions is crucial to mitigating the potential health risks associated with these emerging threats, particularly through a spatiotemporal approach that considers both geographic spread and seasonal variations in risk.

Reviewer #2: Yes

Reviewer #3: (No Response)

**Conclusions**

-Are the conclusions supported by the data presented?

-Are the limitations of analysis clearly described?

-Do the authors discuss how these data can be helpful to advance our understanding of the topic under study?

-Is public health relevance addressed?

Reviewer #1: In lines 409-411, you may want to clarify whether the "developing countries" you mention refer to non-endemic countries or those located in the region near Spain. For example, you could revise the sentence as follows: "It is important to note that while not all infected human cases may be properly reported—especially in developing countries, whether non-endemic or those near Spain—this could lead to a slight underestimation of the actual number of infections." This clarification ensures that the reader understands the specific context of underreporting.

Reviewer #2: Yes

Reviewer #3: (No Response)

**Editorial and Data Presentation Modifications?**

Reviewer #1: While the current title is clear and informative, incorporating key elements from the article could further enhance its specificity and impact. A suggested title revision is: "Assessing the Spatio-Temporal Risk of Aedes-Borne Arboviral Diseases in Non-Endemic Regions: A Data-Driven SIRUVY Model Approach in Northern Spain." This revised title highlights both the methodological approach (SIRUVY model) and the geographic focus, reinforcing the study’s innovative contributions.

Reviewer #2: (No Response)

Reviewer #3: (No Response)

**Summary and General Comments**

Reviewer #1: In lines 9-10 when you mention “Our refined SIRUV human-vector compartmental model incorporates stochastic formulations to capture fluctuations in imported cases, mosquito density, and autochthonous transmission risk - critical factors in outbreak emergence” you could include the SIRUVY model as this “Our refined SIRUV human-vector compartmental model (SIRUVY) incorporates stochastic formulations to capture fluctuations in imported cases, mosquito density, and autochthonous transmission risk - critical factors in outbreak emergence” I believe it is important to explicitly state "SIRUVY" in this way, as it highlights the novelty of the model.

The sentence in lines 26 through 28 “Climate change, combined with rising international human mobility and trade, not only facilitates the proliferation of disease vectors but also drives the introduction of pathogens into naïve areas, increasing interactions between infected individuals and susceptible populations” could be misinterpreted to suggest that vector-borne diseases (VBDs) are directly transmitted between infected individuals and susceptible populations. To clarify that the transmission occurs through vectors, you might consider rewording it like this: "Climate change, combined with rising international human mobility and trade, not only facilitates the proliferation of disease vectors, but also drives the introduction of pathogens into naïve areas, increasing the risk of local vector-mediated transmission to susceptible populations."

In line 33 you could change “arrival of imported cases” for “introduction of imported cases”. If you choose to implement this suggestion and decide on a specific term, I recommend using the same term consistently throughout the article, provided this comment is applicable.

In line 34 “autochthonous cases—locally transmitted infections” sounds redundant. You could use “locally transmitted infections” instead.

On lines 75-76, I would use the word adaptively “Once the relevant data are available, this framework can be adaptively applied to other regions experiencing the introduction of Aedes-borne diseases.”

On lines 96–97, the description of the database is limited. To enhance clarity and reproducibility, consider including additional details such as the total number of reported cases, the period covered by the dataset, and key variables recorded. Additionally, specify whether the cases are confirmed, suspected, or both. If the dataset is publicly available, please indicate whether it is open-source and provide a reference if possible. Similarly, for the entomological data mentioned on lines 97–98, please clarify the nature of the data provided by NEIKER.

On line 106, would it be useful to specify the epidemiological weeks of the years mentioned?

In lines 125-133 when you mention “Additionally, there is a steady external import of new infected cases Y, introduced at a constant arrival rate ◯ϱ.” wouldn’t this be deterministic since new cases are introduced at a constant rate? If there is any stochastic variability in the importation process, consider clarifying whether ◯ϱ represents an expected average rate at a certain point in time, rather than a strictly fixed value. My understanding is that S0, U0 and Y0, serve as initial conditions that shape the model's behavior. However, when applied in different contexts, they can act as adaptive variables rather than fixed parameters, allowing the model to be used in a contextual rather than a purely replicative way. It is key to emphasize that Y0, is crucial for non-endemic regions where outbreaks are seeded by travelers. The contextual number of imported cases will vary over time depending on travel patterns, seasonality, and other external factors. Based on this, these adaptive variables enable the model to be applied in a contextual, time-based, and adaptive manner across different settings, rather than simply replicating it unchanged. It is essential to highlight this point in the article as your audience may not be familiar with this. This would allow to overstress that, as stated in line 63, “deterministic and stochastic models are not designed as direct risk estimators” and as stated in lines 158-159 “However, while this result is derived from a deterministic model, stochasticity is essential to account for the irregular behavior and large fluctuations observed in real disease incidence data”

In lines 173-175 you state “This variability emphasizes the importance of considering stochastic effects in low-endemicity settings, where random fluctuations can greatly determine observed outbreak dynamics and hinder accurate predictions in real-world scenarios.” However, in a dynamic system determination would imply that there is an absolute behavior under certain contextual elements. In that regards you could say “This variability emphasizes the importance of considering stochastic effects in low-endemicity settings, where random fluctuations can greatly influence the determinable nature of observed outbreak dynamics and hinder accurate predictions in real-world scenarios."

While the article discusses imported cases and meteorological conditions, it does not consider other factors that contribute to mosquito proliferation. This is crucial to consider, as these additional factors add to the complexity of the situation and may help in formulating hypotheses or providing a more comprehensive explanation of the dynamics at play. Key factors that should also be taken into account include land use and urbanization patterns, water storage and management practices and socio-economic conditions. These elements can significantly influence mosquito breeding, survival, and disease transmission, making them essential for a more holistic risk assessment.

It is important to document the analysis software and the specific packages used in the study. Please provide references for them to ensure transparency, reproducibility, and clarity for readers who may want to replicate or build upon the analysis.

While the data source is mentioned, it is important to document how the data was processed and prepared for analysis. Providing details on these steps ensures transparency, reproducibility, and a better understanding of the methodological approach used in the study.

While reviewing the article, it appears that the complexity of the situation is addressed through elements of either a systems approach or complexity theory. This is a valuable perspective; however, it would be ideal to emphasize this more explicitly, particularly in relation to complex adaptive systems and emergent phenomena. Highlighting this framework could enhance the understanding of the situation and support the development of future actions based on a more nuanced and dynamic vision, where agents interact in unpredictable ways. However, the key aspect to consider is not predictability itself, but rather how environmental changes facilitate mosquito adaptation. In this context, dengue outbreaks should be understood as emergent phenomena, meaning their occurrence can be anticipated not through precise forecasting, but by identifying early signals that indicate the increasing likelihood of such events. Strengthening surveillance efforts to detect these signals could be instrumental in proactive public health interventions.

Reviewer #2: (No Response)

Reviewer #3: Dr. Guerrero and colleagues have presented an interesting article addressing the epidemiological risk of Aedes-borne arboviral diseases in Northern Spain. While the mathematical modeling is theoretically robust, there are significant concerns regarding the assumptions underlying the model, which, in their current form, compromise the reliability and practical utility of the risk estimates. These assumptions should be thoroughly revised or any statement on the usefulness for risk estimation and applicability considerably tuned down

In particular:

1. The model assumes a constant external importation rate of infected individuals. This assumption is likely unrealistic, as importation rates are expected to vary seasonally, reflecting epidemic dynamics in endemic regions and fluctuations in tourist travel patterns. Furthermore, imported cases may be spatially clustered, particularly in high-density or high-income areas. A time-dependent importation rate might show that most cases arrive during periods when transmission conditions in Northern Spain are unsuitable, or in areas less suitable to disease spread—potentially lowering the actual risk. Notably, the study does not provide any figures or data illustrating the number or distribution of imported cases.

2. The estimation of adult mosquito based on ovitrap data is questionable (see recent review on the AJTMH https://doi.org/10.4269/ajtmh.20-0781 “Its data indicate that it is suitable for assessing the presence of Ae. albopictus at a given site, but not adult abundance”) and raises concerns. It seems to me that, in the end, authors relied on an arbitrary value of kend = 1000 (or its half). Additionally, despite it being a key point in their model, only the three highest positive mosquito egg counts per oviposition stick are reported in the result section. This limited dataset is inadequate to characterize the mosquito population in Northern Spain or to draw meaningful conclusions about epidemiological risk.

3. The parameter values used in the model (as listed in Table 1) are not supported by any cited scientific literature. The absence of justification or references undermines the credibility and reproducibility of the model's outcomes.

PLOS authors have the option to publish the peer review history of their article (what does this mean?). If published, this will include your full peer review and any attached files.

Reviewer #1: No

Reviewer #2: No

Reviewer #3: No

**Figure resubmission:**
---

## [Decision Letter · Decision Letter 1]

7 Jul 2025

Dear Dr. Aguiar,

We are pleased to inform you that your manuscript 'Assessing the Spatio-Temporal Risk of Aedes-Borne Arboviral Diseases in Non-Endemic Regions: The Case of Northern Spain' has been provisionally accepted for publication in PLOS Neglected Tropical Diseases.

Best regards,

Ran Wang, M.D.

Academic Editor

Victoria Brookes

Section Editor

Shaden Kamhawi

co-Editor-in-Chief

Paul Brindley

co-Editor-in-Chief

Reviewer's Responses to Questions

**Key Review Criteria Required for Acceptance?**

**Methods**

-Are the objectives of the study clearly articulated with a clear testable hypothesis stated?

-Is the study design appropriate to address the stated objectives?

-Is the population clearly described and appropriate for the hypothesis being tested?

-Is the sample size sufficient to ensure adequate power to address the hypothesis being tested?

-Were correct statistical analysis used to support conclusions?

-Are there concerns about ethical or regulatory requirements being met?

Reviewer #2: (No Response)

Reviewer #3: (No Response)

**Results**

-Does the analysis presented match the analysis plan?

-Are the results clearly and completely presented?

-Are the figures (Tables, Images) of sufficient quality for clarity?

Reviewer #2: (No Response)

Reviewer #3: (No Response)

**Conclusions**

-Are the conclusions supported by the data presented?

-Are the limitations of analysis clearly described?

-Do the authors discuss how these data can be helpful to advance our understanding of the topic under study?

-Is public health relevance addressed?

Reviewer #2: (No Response)

Reviewer #3: (No Response)

**Editorial and Data Presentation Modifications?**

Reviewer #2: (No Response)

Reviewer #3: (No Response)

**Summary and General Comments**

Reviewer #2: (No Response)

Reviewer #3: I thank the authors for their comprehensive answers and clarifications. i have no further comments

PLOS authors have the option to publish the peer review history of their article (what does this mean?). If published, this will include your full peer review and any attached files.

Reviewer #2: No

Reviewer #3: No

---

## [Editor Report · Acceptance letter]

Dear Dr. Aguiar,

We are delighted to inform you that your manuscript, "Assessing the Spatio-Temporal Risk of *Aedes*-Borne Arboviral Diseases in Non-Endemic Regions: The Case of Northern Spain," has been formally accepted for publication in PLOS Neglected Tropical Diseases.

Best regards,

Shaden Kamhawi

co-Editor-in-Chief

Paul Brindley

co-Editor-in-Chief
